# Psychogenic Non-Epileptic Status as Refractory, Generalized Hypertonic Posturing: Report of Two Adolescents

**DOI:** 10.3390/medicina56100508

**Published:** 2020-09-28

**Authors:** Giangennaro Coppola, Grazia Maria Giovanna Pastorino, Lucia Morcaldi, Floriana D’Onofrio, Francesca Felicia Operto

**Affiliations:** Department of Medicine, Clinic of Child and Adolescent Neuropsychiatry, Surgery and Dentistry, Medical School University of Salerno, 84081 Baronissi, SA, Italy; gcoppola@unisa.it (G.C.); graziapastorino@gmail.com (G.M.G.P.); luciaa-@hotmail.it (L.M.); floriana.donofrio09@gmail.com (F.D.)

**Keywords:** psychogenic non epileptic seizures, hypertonic-posturing, refractory, psychiatric comorbidity, adolescents

## Abstract

Psychogenic non-epileptic seizures (PNES) or dissociative seizures are found under the umbrella headings of functional/dissociative neurological disorders (FND) in psychiatric classifications (DSM-5; ICD-11). PNES are not characterized by any specific ictal or postictal EEG abnormalities. Patients with PNES can present with motor or non-motor symptoms, frequently associated with a change in the level of consciousness. PNES duration is variable, often longer than that of epileptic seizures. Prolonged PNES, sometimes termed PNES status, involve continuous or repetitive events that exceed 30 min. Prolonged PNES are often misdiagnosed as an epileptic event and are often inappropriately treated with high doses of antiseizure drugs. In this report, we describe two adolescent patients who presented with prolonged PNES characterized by generalized hypertonic posturing and low levels of consciousness. Despite multiple presentation to the Emergency department, and multiple normal video-EEG, the patients were misdiagnosed with epilepsy and were inappropriately treated with antiseizure medications. Both patients presented psychiatric comorbidity, consisting of a major depressive disorder, obsessive-compulsive symptoms, social withdrawal, difficulty of social interaction, and anxious-perfectionist personality traits. The episodes of prolonged PNES gradually declined within 18 months in both patients.

## 1. Introduction

Psychogenic non-epileptic seizures (PNES) are paroxysmal events that are commonly seen in adolescents and young adults [1,2]. They have also been named non-epileptic attacks or dissociative seizures and they are found under the umbrella headings of functional/dissociative neurological disorder (FND) in psychiatric classifications [3,4]. PNES account for about 20% of intractable seizure disorders [5] and present a challenge for clinicians in terms of diagnosis and management. Patients with PNES demonstrate high rates of psychiatric comorbidities such as anxiety, depression, and post-traumatic stress disorder [6], as well as a higher risk of mortality that is unrelated to their seizures [7].

PNESs are sometimes misdiagnosed as epileptic seizures, although they can occur in individuals with and without epilepsy [8]. Misdiagnosed PNES patients may go through expensive or unnecessary interventions. Moreover, the life quality of patients with PNES is lower than that of patients with epilepsy [9]. Therefore, a correct diagnosis can reduce medical service use and costs [10] and improve wellbeing and the quality of life.

PNES can also be prolonged-sometimes referred to as a PNES status (Reuber et al., 2003). The term of PNES status has been used when the seizure exceeds 30 min and involves continuous or repetitive paroxysmal clinical events [1]. Importantly, like other PNES, a prolonged PNES is not characterized by any specific ictal or postictal EEG abnormality [11].

As a long lasting and often refractory event, prolonged PNES can particularly bewilder the clinician in the emergency room. Indeed, it can be misdiagnosed as an epileptic seizure, and lead to an overload of antiseizure drugs and to endotracheal and/or pharyngeal intubation maneuvers.

Prolonged PNES in adolescents and young adults are rarely reported [11,12,13]. Furthermore, the clinical symptoms occurring during these events are not always described in detail and/or video documented.

In this report, we describe two cases of adolescent patients who presented a prolonged PNES with overlapping symptoms characterized by a significantly prolonged state of generalized hypertonic posturing with low level of consciousness, which raised diagnostic concerns for treating physicians. A video recording of a typical attack in one of the patients is also available. Informed consent was obtained from both patients and their families.

### 1.1. Case Report 1

#### F.A., A 17-Year-Old Girl

F.A. was born at term after an uneventful pregnancy and delivery. Her birth weight was 3.400 g. She was the only daughter of healthy parents, though her mother suffered from a familial anxiety and mood disorder. Psychomotor development was normal and school attendance was regular with excellent academic achievements.

From the age of ten years, the patient started practicing classical dance and was strongly engaged with it. Soon afterwards, she developed anorexic symptoms for about one year due to her strong desire to be “light” and to perform better. The eating disorder did not require hospitalization, though her parents reported to be very concerned about their daughter’s health condition.

On the other hand, school achievement had always been excellent and at presentation the patient was attending the 4th year of scientific high school. At the beginning of the current school year, the patient had an episode of sudden and persistent hypertonic posturing (up to 2–3 h) that affected the four limbs and trunk, and caused breathing difficulty, mild drooling, revulsion of the eyeballs with unresponsiveness and fall to the ground. This provoked great distress among her classmates and teachers, and led to the immediate call for the emergency medical services.

Once in the emergency room of the local hospital, the patient was misdiagnosed as status epilepticus. She was therefore regularly intubated and treated with a continuous infusion of multiple, high doses of benzodiazepines (midazolam, lorazepam, and diazepam) in combination with antiseizure medications (valproic acid, phenobarbital, and clonazepam). In the subsequent 12 months, the patient underwent more than 15 hospitalizations.

The following paroxysmal events always consisted of an abrupt onset of a generalized hypertonic posturing lasting 2–3 h during which the patient was observed to have difficult breathing and gasping associated with rhythmic abdominal retractions, half-open eyes, unresponsiveness, flashing of the face and neck, and drooling. The clinical symptoms, including the generalized hypertonic posturing, did not respond to continuous infusions of benzodiazepines (Appendix A).

All paroxysmal events occurred during school time and always led to an ambulance call. Over the numerous hospitalizations, serial video-EEG as well as brain Magnetic Resonance Imaging MRI scans were all normal.

With the agreement of the patient and her parents, it was decided to suspend school attendance, obtaining the consent of the teachers to perform scheduled learning assessments.

Subsequently, the paroxysmal events became less frequent. However, the patient exhibited a persisting and stubborn social withdrawal, due to the fear of “seizures” recurrence. A psychodiagnostic assessment, including psychiatric interviews, cognitive profile assessment by means of WAIS scale, and CDI-2 and MMPI-A tests, revealed a major depressive disorder with anxious-perfectionist personality traits and a high intellectual functioning.

The patient could not endure even a minimum school failure and consequently the exclusion from the most deserving group of schoolmates.

After a persistent refusal of psychotherapy which the patient considered to be absolutely “useless”, she presented some improvement with a mood stabilizer associated with a neuroleptic (aripiprazole). After about a year in which intermittent hypertonic attacks occurred (about 1/month), the patient developed frequent headache attacks and again anorexic symptoms with significant weight loss. She currently attends a vocational course and is engaged in a therapeutic program.

### 1.2. Case Report 2

#### B.F., An 18-Years-Old Male

B.F. was born to unrelated parents. His father, aged 60 years, suffered from anxiety-depressive disorder, while his mother, aged 52 years, had a generalized anxiety disorder and depressive traits. He was born at term after an uneventful pregnancy and normal delivery, his birth weight was 3900 g. Psychomotor development was normal and school attendance was regular with good academic achievement.

From the age of 16.5 years, the patient presented, mostly at school, with long-lasting (>2 h), generalized, intense hypertonic body posturing, with “breathing difficulties”, intermittent up-gaze, mild drooling, pallor/flushing of his face.

Once admitted to the emergency department, the paroxysmal events proved to be refractory to high doses of benzodiazepines administered by infusion (diazepam, midazolam, and/or lorazepam). The initial episodes were ineffectively treated with in vein phenobarbital bolus. The patient was regularly intubated with an endopharyngeal tube and treated with an oxygen therapy mask.

The hospital described his attacks as generalized stiffening with hyperextended limbs, feet in equinism, and difficult gasping and breathing.

Brain MRI as well as interictal wake and sleep electroencephalogram EEG recordings were normal. A psychodiagnostic assessment that was carried out when the patient was 17 years old highlighted a major depressive mood disorder with anhedonia, a significant drop in school performance, and a marked social withdrawal. It was documented that the patient was spending most of every day in his bedroom, immersed in his personal computer to navigate the internet. He occasionally frequented a friend with whom he went out in the evening for a few hours. Due to an abnormal circadian sleep-wake rhythm, the patient used to fall asleep in the morning and wake up in the early afternoon.

The withdrawal from school attendance and a drug therapy consisting of a mood stabilizer and an antidepressant Selective Serotonin Reuptake Inhibitors SSRI (venlafaxine) led to the disappearance of the paroxysmal events. The latter, indeed, had only occurred at school. Currently the patient has a more active social life, shows a renewed interest in some of his previous hobbies, and is regularly followed by a psychotherapist.

## 2. Discussion

As described above, the two patients had recurring episodes of prolonged PNES, with a protracted generalized hypertonic posturing (up to 2–3 h) that led to repeated hospitalizations in the emergency department. Both patients were initially misdiagnosed with status epilepticus and were treated with intravenous antiseizure medications, endopharyngeal intubation, and ventilation with an oxygen mask.

Their symptoms were “resistant” to the treatment despite large amounts of different benzodiazepines (midazolam, lorazepam, diazepam) and, at least in the initial episodes, infusional phenobarbital or valproic acid. In both cases, the functional nature of the symptoms, in view of repeated video EEG recordings and a psychodiagnostic assessment, led to a reduction of the pharmacological interventions.

Prolonged PNES in patients under the age of 18 years are reported in a few cases, associated with polymorphic, essentially motor [14,15,16,17], or mixed symptoms- [18,19]-or non-convulsive seizures [20].

The paroxysmal events of our patients appeared to be particularly atypical and bewildering to generalists and psychiatrists in the emergency rooms. Indeed, a persistent state of generalized body hypertonic posturing as the only/predominant symptom associated with respiratory symptoms and unresponsiveness, was misdiagnosed as status epilepticus. Despite repeated normal EEGs, the diagnosis of PNES was not made. Whilst psychogenic non epileptic status is reported in patients both with and without epilepsy [8,18,20,21,22], neither of our patients suffered from epilepsy. Notwithstanding, both patients were treated with high doses of intravenous antiseizure drugs during the repeated hospitalizations in the emergency room. Failure to make the correct diagnosis and the incorrect treatment risked provoking severe iatrogenic complications, including breath depression and induced coma.

Moreover, both patients were unnecessarily intubated, as a result of respiratory suppression from additive doses of depressant drugs. Most of the reported cases of prolonged PNES suggest a remarkable resistance to aggressive doses of depressant drugs, with recurring paroxysmal events in both young and adult patients. This refractoriness, and failure to provide the correct diagnosis and treatment, placed some patients at risk of unnecessary intubation for “airway protection” [16,19,23,24].

As for the age of onset, episodes of prolonged PNES occurred in our two patients at an average age overlapping that previously reported in part of the literature (20.3 years by Reuber et al., 2003; Asadi-Pooya et al., 2013) [1,8].

Indeed, while incidence between juvenile-onset and adult-onset prolonged PNES was quite similar (25% and 23%, respectively by Asadi-Pooya et al., 2013), Reuber et al. (2003) reported that patients with prolonged PNES were younger than patients with PNES of shorter duration-(mean age 20.3 vs. 30.3 years, *p* = 0.001) [1,8].

As for diagnosis, it is of note that the doctors failed to make the correct diagnosis despite normal interictal EEGs. The failure to diagnose could also be explained by the following factors: the seizure type strongly resembled that of generalized tonic-clonic epileptic seizures; emergency room treatment was carried out by different treating physicians unaware of the previous clinical history; the lack of ictal video-EEG recordings. Because the diagnosis was incorrect, so was the treatment plan [25,26,27].

Both adolescents showed symptoms associated with the respiratory system, e.g., difficult breathing and gasping. A recent study of children and adolescents with PNES showed that a significant percentage triggered their PNES via hyperventilation and showed abnormal regulation of PCO_2_ [28]. In neither of the cases in this study was a blood CO_2_ reading taken to assess for hypocapnia that could have been contributing to the dystonic symptom patterns and changes in level of consciousness [29]. Furthermore, breathing issues at the very onset of the paroxysmal events were not asked about with neither patient. It should be noted that there seems to be significant overlap between panic attacks, which involve abnormal breathing patterns, and psychogenic seizures [30]. It is possible that breathing interventions that targeted arousal, a decrease in respiratory rate, and normalization of ventilation and arterial CO_2_ could have been helpful preventing or managing the PNES in the two adolescents described in this report [28].

As for the diagnosis, some authors have suggested that a prolonged PNES can be diagnosed more readily than PNES of shorter duration alone, which, because the latter generally needs a shorter long-term monitoring [20]. We believe this is not always the case if we consider the number of hospitalizations and the use of the emergency room in the clinical history of several patients [16,23,24,25].

In both our patients, the episodes of prolonged PNES gradually declined within 18 months. The paroxysmal events disappeared in both patients upon suspending school attendance.

Soon afterwards, a psychotherapy combined with a psychopharmacological treatment including a mood stabilizer and an antidepressant/neuroleptic treatment were started.

Patient 1 is currently experiencing worrying anorexic behavior with manifest fear of gaining weight and a difficulty in social interaction. Similarly, patient 2 is showing a tendency to social withdrawal with obsessive-compulsive ideation, even if he is slowly recovering academic activities.

The presentations of our patients are consistent with other cases in which a worse outcome is associated with prolonged PNES vs PNES of shorter duration [1] and more frequent hospitalizations in the emergency room [20]. In some cases, the resolution of prolonged PNES occurs within 3.5 years of follow-up [18]. Sometimes, there is comorbid depression and the possibility of a suicidal act or attempt [16,24]. However, recent studies report a full or partial remission in about 80% of cases, following a timely diagnosis and a standardized care pathway for PNES [26,31]. Recently Fobian et al. (2020) reported that a standardized care pathway for PNES management should be multidisciplinary, including cognitive-bahavioral therapy and family support as frontline treatment and pharmacotherapy for comorbid anxiety and depression. Less favourable outcomes were found in children/adolescents who presented with chronic PNES and in those with a chronic comorbid mental health disorder that failed to resolve with treatment [25].

In line with broader literature, our two cases of prolonged PNES are associated with significant psychiatric disorders [26].

In conclusion, we report two cases of adolescents who presented with recurring and prolonged PNES characterized by a persistent and a marked generalized hypertonic posturing with unresponsiveness, leading to heavy antiseizure therapies and occasional endotracheal or endopharyngeal intubation in the emergency room. Despite PNES not being life-threatening even in a status-type situation, the harm in this case may result from a “heavy” pharmacologic treatment. Despite the fact that both patients presented with abnormal breathing, it is noteworthy that no one thought to monitor CO_2_ blood level or even assess respiratory rate.

These paroxysmal events can be easily misdiagnosed by physicians with little knowledge of PNES. A timely video-EEG evaluation will allow a quick diagnosis as well as an early and right treatment in most cases. Multiple presentations are also not needed to make the correct diagnosis. Psychiatric comorbidity in such patients should be carefully considered and addressed, despite the gradual disappearance of the paroxysmal psychogenic motor manifestations. A better knowledge of these symptoms can help to avoid unnecessary and potentially harmful drug therapies and resuscitation maneuvers. It is also mandatory to remember that a longer duration of PNES appears to have detrimental effect on the odds of full remission.

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
