# Peer review of "Psychogenic Non-Epileptic Status as Refractory, Generalized Hypertonic Posturing: Report of Two Adolescents"

_medicina, 2020, doi:10.3390/medicina56100508_

Round 1
Reviewer 1 Report
Overall this reads better than the prior version. I do have ongoing concerns/suggestions as outlined below:
Lines 60-61: for treating physicians rather than in emergency rooms
Line 160: It should be clarified that the intubation happened as a result of respiratory suppression from additive doses of depressant drugs.
Line 188: Clarify what the standardized care pathway is.
Line 192: "Our two cases had significant psychiatric disorders consistent with what has been reported in the literature"
Line 198: You mention that video EEG was performed, but you need to clarify that EEG with concurrent video is necessary to confirm this diagnosis (noting that, as you mentioned in the case reports, the study will show normal electrocortical activity during clinical seizures). You should also provide summary of treatment recommendations, rather than stating the treatment provided for these patients. This is integral to the discussion and to the clarity of the paper.
Overall you need to keep emphasis on the following:
1) this is a rare presentation that mimics status epileptics, the latter being a potentially life-threatening condition that is treated with high doses of seizure medications and often endotracheal intubation to abort seizure activity. This is in contrast to PNES that are not life-threatening even in a status-type situation, and that the harm in these cases results from the treatment with high doses of sedating medications and resulting requirement for airway protection.
2) patients are at risk of morbidity due treatments and delays in diagnosis
3) proper diagnostic steps should be taken early in these cases that are refractory to standard treatments (including clarification of the diagnostic requirements) and
4) early and prompt treatment improves outcomes (including summary of recommended treatment for PNES).
Author Response
Authors’ response:
We thank the reviewer for the comments and the suggestions. We made all required changes, as follows:
Lines 60-61:the words “in emergency room” has been changed to “for treating physicians” (lines 59-60)
Line 160: we clarified that the intubation happened as a result of respiratory suppression from additive doses of depressant drugs in the Discussion section “Moreover, both patients were unnecessarily intubated, as a result of respiratory suppression from additive doses of depressant drugs” (lines 158-159).
Line 188: The follow sentence has been added to the text: “Recently Fobian et al. (2020), reported that a standardized care pathway for PNES management should be multidisciplinary, including cognitive-behavioral therapy and family support as frontline treatment and pharmacotherapy for comorbid anxiety and depression [25]” (lines 198-200).
Line 192: The sentence has been rewritten as follow: “In line with broader literature, our two cases of prolonged PNES are associated with significant psychiatric disorders [30]”(lines 203-204).
Line 198: We clarify that video-EEG is mandatory as diagnostic tool and we provide a summary of treatment recommendations in Discussion section, as follow:
- “These manifestations can be easily misdiagnosed by operators with different resuscitation, neurological or psychiatric skills. A timely video-EEG evaluation will allow a quick diagnosis as well as an early and right treatment in most cases.” (lines 212-214)
- “Recently Fobian et al (2020), reported that a standardized care pathway for PNES management should be multidisciplinary, including cognitive-behavioral therapy and family support as frontline treatment and pharmacotherapy for comorbid anxiety and depression [25]” (lines 198-200)
- “In this regard, both adolescents showed symptoms associated with the respiratory system - difficult breathing and gasping. In neither case, no one bothered to do blood CO2 reading, considering that hypocapnia could have been contributing to the dystonic symptom patterns and changes in level of consciousness [26]. Furthermore, breathing issues at the very onset of the paroxysmal events were not asked for in neither patient. It should be noted that also panic attacks may include abnormal breathing patterns and may be a mask of psychogenic seizures [27]. Check for initial breathing difficulties could have suggested a respiratory control of carbon dioxide including breathing interventions that target arousal, decrease respiratory rate, and normalize ventilation and arterial CO2 [28].” (lines 176-184)
1) We added the follow sentences:
“Despite PNES are not life-threatening even in a status-type situation, the harm in this case may result from a “heavy” pharmacologic treatment. Despite both patients presented with abnormal breathing, it is noteworthy that no one thought to monitor CO2 blood level or even assess respiratory rate.” (lines 208-211)
2) We added the follow sentences:
“Moreover, both patients were unnecessarily intubated, as a result of respiratory suppression from additive doses of depressant drugs”(lines 158-159)
“This refractoriness – and failure to provide the correct diagnosis and treatment - placed some patients at risk of unnecessary intubation for "airway protection" [16,19,23,24]. (lines 161-163)
“Failure to make the correct diagnosis and the incorrect treatment risked provoking severe iatrogenic complications including breath depression and induced coma.” (lines 155-157)
3) We added the follow sentences:
“As for diagnosis, it is of note that the doctors failed to make the correct diagnosis in the first place, despite several interictal EEGs were normal. The failure to diagnose could also be explained by factors including the seizure type strongly resembling that of generalized tonic-clonic epileptic seizures, the first aid carried out by different treating physicians, aneware of the previous clinical hystory, the lack of ictal video-EEG recordings. Because the diagnosis was incorrect so was the treatment plan [25].” (lines 171-176)
“Recently Fobian et al (2020), reported that a standardized care pathway for PNES management should be multidisciplinary, including cognitive-behavioral therapy and family support as frontline treatment and pharmacotherapy for comorbid anxiety and depression [25]” (lines 198-200)
4) We added the follow sentences:
“A timely video-EEG evaluation will allow a quick diagnosis as well as an early and right treatment in most cases. Multiple presentations are also not needed to make the correct diagnosis. Psychiatric comorbidity in such patients should be carefully considered and addressed, despite the gradual disappearance of the paroxysmal psychogenic motor manifestations. A better knowledge of these symptoms can help avoid unnecessary and potentially harmful drug therapies and resuscitation maneuvers. It is also mandatory to remember that a longer duration of PNES illness appears to have detrimental effect on odds of full remission.” (lines 213-219).

Reviewer 2 Report
The manuscript is significantly improved from the original version but it is still needs a second round of revisions.
As noted in the previous review comment I think that the addition of these two cases to the literature would be useful.
Key problems as yet unresolved are:
The “English” in the manuscript is really poor. I have tried to mark some changes in a document. However, the suggested changes do not capture all the problems. I think the authors need to seriously consider asking a colleague who is more fluent in English to check their manuscripts on re-submission. This means that good information is lost. I understand that English is not the authors first language. Nevertheless the English has to be adequate.
Second the authors need to decide on terminology. If they chose the term paroxysmal events then they have to use the term in the whole manuscript. They cannot switch to attacks or other terminology in different sentences. This lack of consistency contributes to the sense that the manuscript has not been properly edited.
The cases clearly highlight the risks to the patient of medical mismanagement—failure to make the proper diagnosis and inappropriate treatment. However medical mismanagement is not adequately raised in the discussion. The authors put a lot of focus on the paroxysmal events being resistant to drug treatment. But they don’t adequately state that the doctors failed to make the correct diagnosis in the first place. They could discuss why this was the case—the factors contributing to failure to misdiagnose—even though the diagnosis of PNES was staring the doctors in the face (normal EEG after normal EEG). Because the diagnosis was incorrect so was the treatment. The authors could make a strong point about the need to make the correct diagnosis and to provide the correct treatment.
Also both adolescents showed symptoms associated with the respiratory system—difficult breathing and gasping. In the conclusion the authors drop this information completely. But abnormal breathing patterns are very common in adolescents with PNES (and adults) (see The respiratory control of carbon dioxide in children and adolescents referred for treatment of psychogenic non-epileptic seizures." European Child and Adolescent Psychiatry 26(10): 1207-1217). Also panic attacks (which also include abnormal breathing patterns) are very common (see Vein AM, Djukova GM, Vorobieva OV (1994) Is panic attack a mask of psychogenic seizures?–a comparative analysis of phenomenology of psychogenic seizures and panic attacks. Funct Neurol 9(3):153–159 or Hendrickson R, Popescu A, Dixit R, Ghearing G, Bagic A (2014) Panic attack symptoms differentiate patients with epilepsy from those with psychogenic nonepileptic spells (PNES). Epilepsy Behav E&B 37:210–214. doi:10.1016/j. yebeh.2014.06.026. Since the breathing issues were key to both presentations I think they need to be discussed.
Likewise the failure of the medical profession to check if the patients were hyperventilating (which would contribute to the change in consciousness) needs to be raised. The patients were intubated and given oxygen but no one bothered to do a blood CO2 reading. No one considered that hypocapnia could have been contributing to the dystonic symptom patterns and changes in level of consciousness (see Engels seminal work in physiology from 1947, Engel, G. L., et al. (1947). "Hyperventilation: Analysis of clinical symptomatology. ." Annals of Internal Medicine 27: 683-704.). Again, because respiratory symptoms were a core part of the presentation I do think they need to be discussed.
There is also a new study the authors might like to include showing that specific treatment for PNES has good outcomes in children (along the other studies they cited).
Fobian, A. D., et al. (2020). "Retraining and control therapy for pediatric psychogenic non-epileptic seizures." Ann Clin Transl Neurol.
The authors may also like to look at the document highlighting some of the problem areas with reference to "English". This might help them get the manuscript into shape.

Author Response
Authors’ response:
We thank the reviewer for the comments and the suggestions. We made all required changes, as follows:
- The English style was widely revised through all the text
- The term “Paroxysmal events” has been used in the whole manuscripts
- The follow sentences was added to the text in the Discussion section: “As for diagnosis, it is of note that the doctors failed to make the correct diagnosis in the first place, despite several interictal EEGs were normal. The failure to diagnose could also be explained by factors including the seizure type strongly resembling that of generalized tonic-clonic epileptic seizures, the first aid carried out by different treating physicians, unaware of the previous clinical hystory, the lack of ictal video-EEG recordings. Because the diagnosis was incorrect so was the treatment plan [25]. In this regard, both adolescents showed symptoms associated with the respiratory system - difficult breathing and gasping. In neither case, no one bothered to do blood CO2 reading, considering that hypocapnia could have been contributing to the dystonic symptom patterns and changes in level of consciousness [26]. Furthermore, breathing issues at the very onset of the paroxysmal events were not asked for in neither patient. It should be noted that also panic attacks may include abnormal breathing patterns and may be a mask of psychogenic seizures [27]. Check for initial breathing difficulties could have suggested a respiratory control of carbon dioxide including breathing interventions that target arousal, decrease respiratory rate, and normalize ventilation and arterial CO2 [28].” (lines 171-184)
- The following references have been added in the text and in the references section:
- Fobian AD , Long DM, Szaflarski JP. Retraining and control therapy for pediatric psychogenic non-epileptic seizures . Ann Clin Transl Neurol 2020 Aug 3;7(8):1410-1419.
- Engel GL, Ferris EB, Logan M. Hyperventilation; analysis of clinical symptomatology . Ann Intern Med 1947 Nov;27(5):683-704
- Hendrickson R, Popescu A, Dixit R, Ghearing G, Bagic A. Panic attack symptoms differentiate patients with epilepsy from those with psychogenic nonepileptic spells (PNES). Epilepsy Behav. 2014 Aug;37:210-4.
- Kozlowska K, Rampersad R, Cruz C, Shah U, Chudleigh C, Soe S, Gill D, Scher S, Carrive P. The respiratory control of carbon dioxide in children and adolescents referred for treatment of psychogenic non-epileptic seizures. Eur Child Adolesc Psychiatry. 2017 Oct;26(10):1207-1217
- Sawchuk T, Buchhalter J, Senft B. Psychogenic nonepileptic seizures in children-Prospective validation of a clinical care pathway & risk factors for treatment outcome. Epilepsy Behav. 2020 Apr;105:106971. doi: 10.1016/j.
- Kozlowska K, Chudleigh C, Cruz C, Lim M, McClure G, Savage B, Shah U, Cook A, Scher S, Carrive P, Gill D.Psychogenic non-epileptic seizures in children and adolescents: Part II - explanations to families, treatment, and group outcomes. Clin Child Psychol Psychiatry. 2018 Jan;23(1):160-176.

Reviewer 3 Report
The authors have improved draft, bus i still recommend intensive editing.
Minor comments: anticonvulsant is not proper term anymore, we say antiseizure drugs, please amend it (lines 54, 141, 157). Line 140: "status epilepticus" not epileptic status, change.
Author Response
Authors’ response:
We thank the reviewer for the comments and the suggestions. We made all required changes, as follows:
The term “anticonvulsant” has been change to “antiseizure” all throught the text
The term “epilepticus status” has been change to “antiseizure” all throught the text
The English style was widely revised through all the text

Round 2
Reviewer 2 Report
Revision 2
This manuscript is much better. However it still have significant language problems—especially with sections that authors added—which need to be fixed. See suggestions below plus an edited version of a paragraph that was very difficult to understand.
The first line of the abstract appears to have a typo. Paroxysmal non-epileptic seizures (PNES) Should be Psychogenic non-epileptic seizures (PNES)
Line 50 delete “PNES patients” and replace with “patients with PNES”
Line 85 delete the exclamation mark. Not clear what it is doing here.
Line 101 the authors need to check if they was serial EEG to read as serial video-EEG
Line 154 the authors need to check if they want to say initially diagnosed or “initially misdiagnosed”
Line 163 is the “to” a typo and needs deletion
Line 165 should manifestations be changed to “events” as in the rest of the manuscript for consistency
Lines 189-203 These lines of new text need further editing. See attached word document
Line 221 need an “and” after the [1]….that is “[1] and more…”
Line 238 RE” Despite PNES are not life-threatening”…
This is not grammatical. It needs to be changed to “Despite PNES not being life-threatening…”
Line 242. RE: “These manifestations…
For consistency the authors need to use paroxysmal events
Line 242-243 RE: “by operators with different resuscitation, 242 neurological or psychiatric skills”.
This line does not make sense. Do the authors mean physicians with little knowledge of PNES?
Line 249 RE : duration of PNES illness
The word illness is not needed here and can be dropped.
Please look also at suggestions for paragraph in attached file.

Author Response
Salerno, September 25, 2020
To Editor-in-Chief
Brain Sciences
Dear Editor,
Thank you for considering our manuscript for publication in Brain Sciences.
We greatly appreciated the reviewer’s comments and suggestions. The manuscript has been modified, according to the reviewer’s comments and remarks. We have italicized the comments and addressed each of them in a point-by-point reply.
We hope that the manuscript is now acceptable for publication in Brain Sciences.
Looking forward to hearing from you,
Francesca Felicia Operto
Child and Adolescent Neuropsychiatry Unit, Department of Medicine, Surgery and Dentistry, University of Salerno. Via Salvator Allende, 84081 Baronissi (SA) - Italy.
Phone: +39 0828672578; +39 3471735041:
email: opertofrancesca@gmail.com.
COMMENTS TO THE AUTHORS
REVIEWER: 2
Reviewer comment:
This manuscript is much better. However it still have significant language problems—especially with sections that authors added—which need to be fixed. See suggestions below plus an edited version of a paragraph that was very difficult to understand.
The first line of the abstract appears to have a typo. Paroxysmal non-epileptic seizures (PNES) Should be Psychogenic non-epileptic seizures (PNES)
Line 50 delete “PNES patients” and replace with “patients with PNES”
Line 85 delete the exclamation mark. Not clear what it is doing here.
Line 101 the authors need to check if they was serial EEG to read as serial video-EEG
Line 154 the authors need to check if they want to say initially diagnosed or “initially misdiagnosed”
Line 163 is the “to” a typo and needs deletion
Line 165 should manifestations be changed to “events” as in the rest of the manuscript for consistency
Lines 189-203 These lines of new text need further editing. See attached word document
Line 221 need an “and” after the [1]….that is “[1] and more…”
Line 238 RE” Despite PNES are not life-threatening”…
This is not grammatical. It needs to be changed to “Despite PNES not being life-threatening…”
Line 242. RE: “These manifestations…
For consistency the authors need to use paroxysmal events
Line 242-243 RE: “by operators with different resuscitation, 242 neurological or psychiatric skills”.
This line does not make sense. Do the authors mean physicians with little knowledge of PNES?
Line 249 RE : duration of PNES illness. The word illness is not needed here and can be dropped.
Please look also at suggestions for paragraph in attached file.
Authors’ response:
We thank the reviewer for the comments and the suggestions. We have made all the changes requested by the reviewer, which are in red in the tracked version. We have also included all the changes made by the reviewer in the added paragraph.
This manuscript is a resubmission of an earlier submission. The following is a list of the peer review reports and author responses from that submission.
Round 1
Reviewer 1 Report
The diagnosis and treatment of PNES is a topic of consideration interest worldwide. The cases described in the article would make a nice contribution to the literature. Prolonged PNES are common and they are often treated inappropriately as the authors describe.
However the article, in its current form, has some problems, both English-wise and some other problems that also need to be addressed.
Also the authors seem to reference quite old literature. But newer studies about treatment are not referenced. For example, Terry et al. (2020); Sawchuk, T., et al. (2020); Kozlowska et al. 2018 (full references are below).
There is also no acknowledgement to the adolescents and their families. Presumably the authors have consent to write up the case studies and to post the video on line. However consent issues are not mentioned explicitly, so I do not know what the consent status of the paper is.
Abstract line 13-15
RE: Paroxysmal non-epileptic seizures (PNES) or dissociative seizures are found under the umbrella headings of dissociative and conversion disorders in psychiatric classifications (DSM-5; 14 ICD-11).
Most clinicians working in the field today are using the term functional neurological disorder/functional neurological symptom disorder (FND) rather than conversion disorder. Many say, functional neurological (conversion) disorder. I think the authors might like to consider doing the same (if they want to be more current).
Line 15-17 RE: They can appear with multiple symptoms with a motor or non-motor prevalence associated with neurovegetative disorders and low level of consciousness. Their duration is variable, often longer than that of epileptic seizures.
This sentence is not grammatical. The first word “they” refers to PNES. PNES are not a person and they cannot appear with multiple symptoms. I think the authors want to say that Patients with PNES can present with multiple symptoms…..
I do not understand what neurovegetative disorders means here? The authors are talking about PNES here, what does neurovegetative disorders refer to? Do the authors mean comorbid anxiety and depression or something else?
Line 18: PNES status
This is not a term that I have ever seen used in the current/recent literature. Epileptic status always refers to epileptic seizures.
Maybe it is used in Italian?
I see from the reference list that Palkanis 2000 used psychogenic status epilepticus and Taxhorn 2000 used pseudostatus epilepticus, Jagoda (1995) used psychogenic status epilepticus, papavasiliou (2004) used psychogenic status epilepticus. These are now quite old papers.
It seems like the authors are introducing yet another term PNES status. This adds yet another term to an already confused field.
I personally do not see the addition of more terms as useful. It just increases the number of words that are used for the same phenomenon. It also conveys the idea that prolonged PNES are something different—an entity of its own. In fact prolonged PNES are very common and part of any cohort of PNES.
From my perspective, I would be more comfortable if the authors changed the terminology i into ‘Prolonged PNES’.
Also the term “PNES status” raises the question of the quality of treatment. Were the PNES so prolonged for such a long period of time because the clinicians involved did not implement appropriate treatment? The diagnosis was not made promptly, a good explanation was not given to the family, the family rejected the diagnosis? I think the cases both raise the issue as to why the diagnosis took so long to make and the treatment intervention took a long time to implement.
Line 23 RE: a kind of PNES status
Again this terminology is problematic.
Line 24 RE: to our knowledge not previously described in the literature
The authors in the following article described a PNES episode of 4 hours duration.
Kozlowska, K., et al. (2016). "The body comes to family therapy: Treatment of a school-aged boy with hyperventilation-induced non-epileptic seizures." Clinical Child Psychology and Psychiatry 21(4): 669-685.
Line 36 RE: Psychogenic non-epileptic seizures (PNES) are not rare paroxysmal events, particularly in 36 adolescents and young adults
Are not rare paroxysmal events if not good English.
The authors could say: Psychogenic non-epileptic seizures (PNES) are paroxysmal events that are commonly seen in adolescents and young adults.
Line: 37-39 RE:
They have also been named as non-epileptic attacks or dissociative seizures and they are found under the umbrella headings of dissociative and conversion disorders in psychiatric classifications
Delete the as – not needed.
See comment about terminology in abstract.
Line 44 RE: epileptic events
This should be epileptic seizures I think.
Line 49-50: RE: While rarely, a PNES can also manifest as a status. A PNES status occurs when the following criteria are met: 49 the duration of the seizure exceeds 30 minutes and involves continuous or repetitive paroxysmal clinical events.
The authors give a definition but it is not referenced. Is it their own or has someone else said this? Where is it from. In our own clinical practice we use 60 min as a cut-off after which the parents may like to bring the child with PNES into hospital. So clearly practice differs.
Line 53 RE epileptic symptom
Surely the authors mean epileptic seizure
Line 54 and sometimes also admission to ICU something I have seen very often.
Line 55 RE PNES status reports
As noted previously I struggle with the terminology.
Line 59 RE: who presented a kind of PNES status-
See previous comment. Also “a kind of” does not sound very definite.
Line 74 RE: school results have always been particularly bright
An English issue here.
A child can be bright but School results cannot. Change to make grammatical.
Line 92 RE: scans.resulted to be normal.
English problem and typo
I think the authors mean “scans were all normal”.
Line 104 RE: psychotherapeutic help
Do the authors mean psychotherapy or something else?
Line 105 RE: some margins of improvement
English problem here—this is not an English term. I think the authors mean “some improvement” or some “limited” improvement”
Line 108-109 RE: is following a psychotherapeutic treatment.
English problem
Do the authors mean? “ is engaged in a therapeutic program” ???
Line 118
I notice that both patients have breathing difficulties suggesting that they may have triggered their PNES via hyperventilation (HV).
The video shows the patient breathing quite quickly—and a disordered pattern of breathing—and I wonder whether the PCO2 was ever measured during events to see if HV was a trigger. In a perfectionist population with comorbid anxiety this is quite common.
Hyperventilation is an important factor/trigger for PNES. See Sawchuk, T., et al. (2020) for information about HV (and also references in the reference list pertaining to HV)>
Line 31-32 RE: through non-targeted and inconclusive activities
English problem.
I don’t understand what this means.
Line 142-143 RE: Both of them were initially diagnosed with epileptic status
I think the authors mean misdiagnosed. That is, wrongly diagnosed.
Line 147-148 In both cases, a supervened suspect of the psychogenic nature of the symptoms after a variable number of attacks led to a reduction of the pharmacological interventions.
This is not very clear. Surely the diagnosis of PNES was made after video EEG or repeated video EEG?
Line 150 RE: Psychogenic non epileptic status
As noted before I have difficulty with this terminology. PNES lasting more than 30 minutes are very common. I don’t think a new term is needed for them.
Line 157-158 RE: Furthermore, to our knowledge PNES-status with this symptomatology has never been reported so far.
From memory there is an old picture/drawing of a girl arching into a PNES from Charcot’s time. Maybe the authors need to find it and refer to it and say that historically this is semiology that has recurred over history. I will try and paste it into the review. I am not sure which text book it is from.
Line 178- 181 RE: Moreover, Dworetzky et al., (2006) have argued that if it is true that diagnosis is quickly made 178 by means of long-term monitoring, it is also true that a PNES status can be continuously 179 misdiagnosed as epileptic seizure before being studied and mistreated with anticonvulsant drugs 180 accordingly.
English problem. I do not understand what the authors are trying to say.
Surely if one does vEEG and makes a quick diagnosis then one can get on with treatment. For treatment articles see:
Terry, D., et al. (2020). "Outcomes in Children and Adolescents With Psychogenic Nonepileptic Events Using a Multidisciplinary Clinic Approach." Journal of Child Neurology: 883073820939400.
Sawchuk, T., et al. (2020). "Psychogenic nonepileptic seizures in children-Prospective validation of a clinical care pathway & risk factors for treatment outcome." Epilepsy & Behavior 105: 106971.
Kozlowska, K., et al. (2018). "Psychogenic non-epileptic seizures in children and adolescents: Part II - explanations to families, treatment, and group outcomes." Clinical Child Psychology and Psychiatry 23(1): 160-176.
Line 182-182 RE: In both our patients, the episodes of PNES status gradually declined within 18 months after a 182 correct diagnosis was given and antiseizure medications were withdrawn.
Surely treatment the comorbid anxiety might have helped?
This aspect is mentioned a few sentence later but surely it needs to be up here also. Just the diagnosis and just the cessation of epileptic drugs did not fix the problem.
Line 183-184 RE: The paroxysmal episodes 183 have definitively disappeared in both patients upon suspending attendance.
What does this mean? Attendance at school?
Interesting. In other treatment programs continuation at school is a key goal and treatment outcome. See treatment articles above.
Line 192 RE: In some cases the resolution of status episodes occurs within 3.5 years of follow-up
What about newer outcome studies that assessment and treat kids proactively from the start?
Terry, D., et al. (2020). "Outcomes in Children and Adolescents With Psychogenic Nonepileptic Events Using a Multidisciplinary Clinic Approach." Journal of Child Neurology: 883073820939400.
Sawchuk, T., et al. (2020). "Psychogenic nonepileptic seizures in children-Prospective validation of a clinical care pathway & risk factors for treatment outcome." Epilepsy & Behavior 105: 106971.
Kozlowska, K., et al. (2018). "Psychogenic non-epileptic seizures in children and adolescents: Part II - explanations to families, treatment, and group outcomes." Clinical Child Psychology and Psychiatry 23(1): 160-176.
Line 194 RE: In addition, our two cases seem to confirm that PNES status compared to PNES alone
Data from other studies suggest a very high comorbidity with all PNES.
I think this division is artificial.
See
Sawchuk, T., et al. (2020) and other articles suggested before
Line 195-196 RE: developing personality traits such as 195 major depressive disorder, anxiety, obsessive-compulsive disorder, post-traumatic stress disorder, 196 social withdrawal, difficulty of interaction, anxious-perfectionist traits
A problem here: major depressive disorder, anxiety, obsessive-compulsive disorder, post-traumatic stress disorder are not personality traits.
Line 198-2 RE:
In this regard, it is interesting to report that there is a possible overlapping between the brain 198 areas involved in the psychiatric disorders presented by our two patients, namely the hyperactivity 199 of the anterior cingulate cortex in the obsessive-compulsive disorder [26], the microglial activation of 200 the anterior cingulate cortex in the major depressive disorder [27], the large involvement of the 201 medial orbitofrontal cortex and right insula in anorexia nervosa [28], and an abnormal reactivity of 202 the anterior cingulate cortex and insula as a part of a more complex stress response circuitry in 203 subjects with psychogenic non-epileptic seizures [29,30].
It seems to me that most psychiatric disorders involve the brain stress circuitry. I think that this paragraph makes huge leaps which is it better not to make. For example, the Ristic study is in adults: How do the authors know that children/youth have the same changes? Other MRI studies in FND had shown that structural findings in children and adults may not be the same.
Here the authors are trying to connect FND with everything because of the presentation of two patients!! I think this paragraph over-reaches and needs to be dropped. The authors present two very nice cases. This do not need to over-reach to make the article helpful.
Line 209-213
I think the conclusion needs to say something about quick diagnosis via video EEG and getting onto the right path of treatment much earlier. Multiple presentations are not needed to make the correct diagnosis. This is an important element to highlight I think.
Reviewer 2 Report
The draft might be potentially interesting for community but needs improvement. It is very scruffy. Langauge editing necessary. A lot of errors, typos, inconsistencies in using abbrevations.
Reviewer 3 Report
I think the authors should consider the overall goals of the manuscript/case report. At present it is cloudy and hard to distinguish the important points. Are they wanting to bring to the attention of the reader this unique semiology of psychogenic status, the underlying and potential psychiatric co-morbidities or the young age of the affected patients?
There are some other issues-
1) The lack of discussion of formal diagnosis of PNES in the first patient. 2) Consideration of proper treatment of status epileptics as maltreatment (This can be re-worded to state what I assume the authors intended- that proper treatment of presumed status epileptic was applied to patients who were not having epileptic seizures, resulting in potential detriment to the patients). 3) Poor syntax.